

# Phylogenetic analysis of higher-level relationships within Hydroidolina (Cnidaria: Hydrozoa) using mitochondrial genome data and insight into their mitochondrial transcription

Ehsan Kayal[1], Bastian Bentlage[1], Paulyn Cartwright[2], Angel A. Yanagihara[3], Dhugal J. Lindsay[4], Russell R. Hopcroft[5] and Allen G. Collins[1,6]

[1] Department of Invertebrate Zoology, Smithsonian Institution, Washington, DC, USA
[2] Department of Ecology and Evolutionary Biology, University of Kansas, Lawrence, KS, USA
[3] Department of Tropical Medicine, Medical Microbiology and Pharmacology, John A. Burns School of Medicine, University of Hawaii at Manoa, Honolulu, HI, USA
[4] Japan Agency for Marine-Earth Science and Technology (JAMSTEC), Yokosuka, Japan
[5] Institute of Marine Science, University of Alaska Fairbanks, Fairbanks, AK, USA
[6] National Systematics Laboratory of NOAA's Fisheries Service, National Museum of Natural History, Washington, DC, USA

Corresponding author
Ehsan Kayal, kayale@si.edu

## ABSTRACT

Hydrozoans display the most morphological diversity within the phylum Cnidaria. While recent molecular studies have provided some insights into their evolutionary history, sister group relationships remain mostly unresolved, particularly at mid-taxonomic levels. Specifically, within Hydroidolina, the most speciose hydrozoan subclass, the relationships and sometimes integrity of orders are highly unsettled. Here we obtained the near complete mitochondrial sequence of twenty-six hydroidolinan hydrozoan species from a range of sources (DNA and RNA-seq data, long-range PCR). Our analyses confirm previous inference of the evolution of mtDNA in Hydrozoa while introducing a novel genome organization. Using RNA-seq data, we propose a mechanism for the expression of mitochondrial mRNA in Hydroidolina that can be extrapolated to the other medusozoan taxa. Phylogenetic analyses using the full set of mitochondrial gene sequences provide some insights into the order-level relationships within Hydroidolina, including siphonophores as the first diverging clade, a well-supported clade comprised of Leptothecata-Filifera III–IV, and a second clade comprised of Aplanulata-Capitata *s.s.*-Filifera I–II. Finally, we describe our relatively inexpensive and accessible multiplexing strategy to sequence long-range PCR amplicons that can be adapted to most high-throughput sequencing platforms.

## INTRODUCTION

Cnidaria (corals, anemones, jellyfish, hydroids) is a phylum of relatively simple aquatic animals characterized by the presence of a specific cell type, the cnidocyte, which harbors a highly specialized cellular organelle, the cnidocyst. Cnidaria encompasses five recognized classes (*Daly et al., 2007*): Anthozoa (stony corals, sea anemones, tube anemones, soft corals and gorgonians), Cubozoa (box jellyfish), Hydrozoa (hydroids, hydromedusae and siphonophores), Scyphozoa (the so-called true jellyfish), and Staurozoa (stalked jellyfish). Non-anthozoan cnidarians are united in the clade Medusozoa (*Collins, 2002*), whose members typically display a metagenetic life cycle consisting of planula larva, sessile polyp and free-swimming medusa, not all of which may be present in the life cycle of a given species. Within Medusozoa, Hydrozoa represents, to many measures, the most diverse class. Hydrozoa encompasses over 90% of medusozoan species (*Daly et al., 2007*), and so it is perhaps unsurprising that life cycle variation, as well as disparity of medusae, polyps, and colonies within this class far exceeds what is observed within Cubozoa, Scyphozoa or Staurozoa. An important and necessary step in understanding the evolution of the remarkable biodiversity present within Hydrozoa is a robust hypothesis of the phylogenetic relationships among its component taxa.

Recent work based on nuclear ribosomal sequences (*Collins, 2002*; *Collins et al., 2006*; *Collins et al., 2008*; *Cartwright et al., 2008*) and complete mitochondrial genome sequences (*Kayal et al., 2013*) shows that Hydrozoa consists of two main clades, Trachylina and Hydroidolina. Monophyly of the latter is also supported by phylogenetic analyses of life history and anatomical features (*Marques & Collins, 2004*). Trachylina is relatively poor in terms of species richness, containing roughly 150 species in four orders: Limnomedusae, Trachymedusae, Narcomedusae and Actinulida (*Collins et al., 2008*). The remainder of the approximately 3,350 species of hydrozoans (*Daly et al., 2007*) that make up the clade Hydroidolina, are classified in three orders: Anthoathecata, Leptothecata and Siphonophora (*Schuchert, 2015*; Hydroidolina. Accessed through: *Schuchert (2015)* World Hydrozoa database at http://www.marinespecies.org/hydrozoa/aphia.php?p=taxdetails&id=19494 on 2015-07-09). Hydroidolina comprises almost all hydrozoans whose life cycle includes a benthic polypoid or hydroid stage (the exception being Limnomedusae, which is part of Trachylina). Colonial hydroid stages within Hydroidolina, especially siphonophores, tend to have greater functional specialization between zooids than other colonial members of Cnidaria (*Hyman, 1940*; *Dunn, Pugh & Haddock, 2005*; *Dunn, 2009*; *Cartwright & Nawrocki, 2010*).

Two of the three presently recognized orders within Hydroidolina (Anthoathecata, Leptothecata and Siphonophorae) have strong support for their monophyly: Leptothecata whose constituent species' hydroid phase contains a theca (with a few exceptions) and whose medusae (when present) have gonads along the radial canals (*Cartwright et al., 2008*; *Leclère et al., 2009*), and Siphonophorae, pelagic animals with a remarkable level of colony organization (*Dunn, Pugh & Haddock, 2005*; *Cartwright et al., 2008*; *Dunn, 2009*). In contrast, no phylogenetic analysis has provided support for the monophyly of Anthoathecata. Anthoathecata contains those species that lack a theca during the hydroid phase and

whose medusae (when present) usually bear gonads on the manubrium (*Cartwright et al., 2008*). Yet, the absence of thecae can easily be interpreted as plesiomorphic (*Cartwright & Nawrocki, 2010*). Even though there has been no support for the monophyly of Anthoathecata, several likely clades have been identified within this taxonomic group. Aplanulata, a group consisting of hydrozoans that lack a ciliated planula stage, was introduced recently (*Collins et al., 2005*) and its monophyly supported in subsequent studies (*Collins et al., 2006*; *Cartwright et al., 2008*; *Kayal et al., 2013*; *Nawrocki et al., 2013*). Aplanulata contains a portion of the families whose species, in the hydroid stage, possess capitate tentacles. In the past, all hydrozoan species possessing capitate tentacles have been united within the anthoathecate suborder Capitata. However, the taxon has become restricted to a well-supported clade of non-Aplanulata species with capitate tentacles (*Collins et al., 2005*; *Nawrocki, Schuchert & Cartwright, 2010*), referred to as Capitata *sensu stricto* by *Cartwright et al. (2008)*. The status of the anthoathecate suborder Filifera, containing species whose hydroid stage has tentacles with more or less uniform distribution of nematocysts (filiform), is even more complex, with no less than four putative clades with various levels of support recognized (*Cartwright et al., 2008*; *Cartwright & Nawrocki, 2010*).

Despite the recognition of several possible and likely clades within Hydroidolina, phylogenetic analyses have thus far suffered from low support for deep nodes representing the relationships among them. Filifera has never been recovered as a monophyletic group in any explicit phylogenetic analysis, nor has there been support for relationships among the filiferan clades, Capitata, Aplanulata, Leptothecata and Siphonophora (*Collins, 2002*; *Collins et al., 2006*; *Collins et al., 2008*; *Cartwright et al., 2008*; *Cartwright & Nawrocki, 2010*). Lack of resolution among the deep nodes of Hydroidolina hinders our understanding of their evolution. Indeed, a recent review (*Cartwright & Nawrocki, 2010*) highlighted the complexity of morphological characters in the evolutionary history of Hydrozoa and lamented the current lack of resolution of hydroidolinan phylogeny, particularly at ordinal and subordinal levels, which prevents a better understanding of life cycle evolution within this class.

Recent technological advances have allowed us to target the nearly complete mtDNA instead of the single-locus approaches, including barcoding, often used in systematics and biodiversity studies (*Dettai et al., 2012*). The small size and circular nature of the majority of animal mtDNAs make them accessible for low-budget taxonomic studies, given the availability of simple and inexpensive protocols. It is now possible to amplify the complete mtDNA using long-range PCR (*Burger et al., 2007*), which combined with novel high-throughput sequencing technologies, provide access to mitogenomic data for groups considered "difficult-to-sequence" at very low cost and effort (*Kayal et al., 2012*; *Briscoe et al., 2013*; *Foox et al., 2015*). To date, the mtDNA of 188 non-bilaterian animals has been sequenced, out of which 124 are cnidarians, mostly anthozoans.

Medusozoan mtDNA sequencing presents a unique challenge in that all medusozoans possess linear mitochondrial genomes (*Kayal et al., 2012*). Sequencing complete linear chromosomes using traditional long-PCR approach requires knowledge of genome organization, particularly the genes at the ends of the linear molecules. Specifically, studies
have suggested that the mitochondrial genome in medusozoan cnidarians can be mono-, bi-, or octo-chromosomal (*Ender & Schierwater, 2003*; *Voigt, Erpenbeck & Wörheide, 2008*; *Kayal & Lavrov, 2008*; *Park et al., 2012*; *Zou et al., 2012*; *Kayal et al., 2012*; *Smith et al., 2012*). Interestingly, the linearization of medusozoan mtDNA appears to coincide with relative stability in the gene organization of most medusozoans (*Kayal et al., 2012*), which facilitates designing protocols for amplification and sequencing most of the coding regions of the mitochondrial chromosome(s).

We present an analysis of nearly-complete mitochondrial genome sequences from a diverse set of hydrozoan taxa in an effort to better understand the relationships within Hydroidolina. Specifically, we describe twenty-six novel, nearly-complete mitochondrial genomes from several hydrozoan orders. We first analyzed the composition and gene order of these mitochondrial genomes. We then used RNA-seq data to infer some of the mechanisms involved in mitochondrial gene expression. Finally, we used both the nucleotide and amino acid sequence data to reconstruct the evolutionary history of hydrozoans, focusing on the thus far intractable relationships within Hydroidolina.

## MATERIAL AND METHODS

### Taxon sampling

We sampled species from both hydrozoan subclasses, Trachylina (three species) and Hydroidolina (twenty-three species), maximizing the coverage of hydrozoan diversity by sampling at least one species from all the currently recognized hydroidolinan clades that correspond to the orders/suborders Aplanulata, Capitata *s.s.*, Filifera I–IV, Leptothecata, Siphonophorae (Table 1). We acquired all publicly available medusozoan mitochondrial genomes through Genbank, including nineteen non-hydrozoans used as outgroup taxa (Table 1).

### Obtaining nearly-complete mitochondrial genomes

We followed the protocol described in a previous study (*Kayal et al., 2012*) to amplify the nearly-complete mitochondrial DNA (mtDNA) of sixteen hydrozoan species. This protocol exploits the relative conservation of the gene organization within Hydrozoa to amplify the nearly-complete mtDNA in one or two pieces via long-range PCR. First, we used conserved metazoan primers to amplify and sequence regions of *cox1* and *rns* genes in all sampled taxa. For the two species of Trachylina, we also amplified and sequenced regions of *cob* and *rnl*. Finally, for several species *rns* was difficult to amplify and we sequenced *nad5* instead. We then designed species-specific and conserved primers for long-range PCR amplification as described in *Kayal et al. (2012)*. We amplified the nearly complete mtDNA (encompassing most coding regions) in one or two contigs using Ranger Taq (Bioline, London, UK) with a combination of one, two or three sets of primers (see Table S1 for the list of primers and lengths of long PCR amplicons per species). Long amplicons were visualized on an Agarose gel, when necessary multiple amplicons were pooled for each individual specimen, and sheared to the appropriate size-range using a Q800R sonicator (QSONICA, Newton, Connecticut, USA). Sheared amplicons

**Table 1** List of samples used in this study.

| Clade | | Species | Voucher | Accession[a] | Note[b] |
|---|---|---|---|---|---|
| Aplanulata | | *Boreohydra simplex* | Borehydra20100904.3 | KT809334 | Long-range PCR |
| | | *Ectopleura larynx* | | JN700938 | |
| | | *Ectopleura larynx*[*] | | LN901195 | RNA-seq; SRR923510 |
| | | *Euphysa aurata* | GR10-145.2 | KT809330 | Long-range PCR |
| | | *Hydra oligactis* | | NC_010214 | |
| | | *Hydra magnipapillata* | | NC_011220–NC_011221 | |
| | | *Hydra vulgaris* | | HM369413–HM369414 | |
| | | *Plotocnide borealis* | RU087.1 | KT809334 | Long-range PCR |
| Capitata | | *Cladonema pacificum* | | KT809323 | DNA-seq; unpublished raw reads |
| | | *Millepora platyphylla* | | JN700943 | Old Millepora EK-2011 |
| | | *Pennaria disticha* | | JN700950 | |
| | | *Sarsia tubulosa* | RU053 | KT809333 | Long-range PCR |
| Filifera | IV | *Catablema vesicarium* | RU006 | KT809324 | Long-range PCR |
| | III | *Clava multicornis* | | NC_016465 | |
| | I | *Eudendrium capillare* | PS101 | KT809336 | Long-range PCR |
| | IV | *Halitholus cirratus* | GR10-115 | KT809337 | Long-range PCR |
| | III | *Hydractinia polyclina* | | LN901196 | RNA-seq; SRR923509 |
| | III | *Hydractinia symbiolongicarpus* | | LN901197 | RNA-seq; SRR1174275 & SRR1174698 |
| | IV | *Leuckartiara octona* | PS487 | KT809325 | Long-range PCR |
| | IV | *Nemopsis bachei* | | JN700947 | |
| | III | *Podocoryna carnea* | | LN901210 | RNA-seq; SRR1796518 |
| | II | *Proboscidactyla flavicirrata* | PS139 | KT809319, KT809329 | Long-range PCR |
| | IV | *Rathkea octopunctata* | RU008 | HT809320 | Long-range PCR |
| Leptothecata | | *Laomedea flexuosa* | | NC_016463 | |
| | | *Melicertum octocostatum* | RU082 | KT809321 | Long-range PCR |
| | | *Mitrocomella polydiademata* | RU060 | KT809332 | Long-range PCR |
| | | *Ptychogena lactea* | GR10-152.1 | KT809322 | Long-range PCR |
| | | *Tiaropsis multicirrata* | GR10-053.1 | KT809326 | Long-range PCR |
| Siphonophorae | | *Nanomia bijuga* | | LN901198–LN901208 | RNA-seq; SRR871527 |
| | | *Physalia physalis* | | LN901209 | RNA-seq; SRR871528 |
| | | *Physalia physalis* | Angel | KT809328 | RNA-seq; unpublished raw reads |
| | | *Rhizophysa eysenhardti* | DLSI230 | KT809335 | Long-range PCR |
| Trachylina | | *Craspedacusta sowerbyi* | | NC_018537 | |
| | | *Craspedacusta sowerbyi* | | LN901194 | RNA-seq; SRR923472 |
| | | *Cubaia aphrodite* | | NC_016467 | |
| | | *Geryonia proboscidalis* | BCS32a | KT809331 | Long-range PCR |
| | | *Liriope tetraphylla* | | KT809327 | DNA-seq; unpublished raw reads |
| Discomedusae | | *Aurelia aurita* | | NC_008446 | Shao et al. (2006) |
| | | *Aurelia aurita* | | HQ694729 | *Park et al. (2012)* |
| | | *Cassiopea andromeda* | | JN700934 | |
| | | *Cassiopea frondosa* | | NC_016466 | |

| Clade | Species | Voucher | Accession[a] | Note[b] |
|-------|---------|---------|--------------|---------|
| | *Catostylus mosaicus* | | JN700940 | |
| | *Chrysaora quinquecirrha* | | HQ694730 | |
| | *Chrysaora sp. EK-2011* | | JN700941 | |
| | *Cyanea capillata* | | JN700937 | |
| | *Rhizostoma pulmo* | | JN700987–JN700988 | |
| | *Pelagia noctiluca* | | JN700949 | |
| Coronatae | *Linuche unguiculata* | | JN700939 | |
| Staurozoa | *Craterolophus convolvulus* | | JN700975–JN700976 | |
| | *Haliclystus sanjuanensis* | | JN700944 | |
| | *Lucernaria janetae* | | JN700946 | |
| Cubozoa | *Alatina moseri* | | JN642330–JN642344 | |
| | *Carukia barnesi* | | JN700959–JN700962 | |
| | *Carybdea xaymacana* | | JN700977–JN700983 | |
| | *Chironex fleckeri* | | JN700963–JN700968 | |
| | *Chiropsalmus quadrumanus* | | JN700969–JN700974 | |

**Notes.**

[*] Not included in phylogenetic analyses.

[a] Accession name correspond to KT for Genbank and LN for the European Nucleotide Archive (ENA).

[b] SRR codes are GenBank Archive numbers of the DNA-seq and RNA-seq runs used in this study.

were processed for multiplexed double-tagged library preparation for Illumina (100 bp single-end) or Ion Torrent (200 bp single-end) sequencing using custom protocols (see Supplemental Information for detailed protocols). Sequencing was performed either on one lane of Illumina HiSeq2000 platform (Illumina, San Diego, California, USA) at the Genomics Core Lab of the University of Alabama or using one 316 v.1 chip on the Ion Torrent Personal Genome Machine Ion platform (PGM; Life Technologies, Carlsbad, California, USA) at the Laboratories of Analytical Biology of the Smithsonian National Museum of Natural History.

## Sequence assembly and annotation

Sequence reads were sorted per taxon by index and barcode using the Galaxy Barcode Splitter from the Galaxy platform (*Giardine et al., 2005*; *Blankenberg et al., 2010*; *Goecks, Nekrutenko & Taylor, 2010*) and Geneious v.7 (*Kearse et al., 2012*), respectively. Reads were trimmed and the barcode removed using Geneious before proceeding to assembly using the built-in overlap-layout-consensus assembler of Geneious v.7 and a modified version of MITObim v.1.7 (*Hahn, Bachmann & Chevreux, 2013*). Then, we used these consensus sequences as backbones to map the sorted and end-trimmed raw reads using both MIRA v.4 (*Chevreux, Wetter & Suhai, 1999*) and the built-in Geneious mapping plug-in. The final contigs covered the nearly complete mtDNAs as expected from long-range PCR amplifications.

We also probed several large sequence libraries: DNA-seq libraries obtained from a specimen of *Liriope tetraphylla* and two non-clonal specimens of *Cladonema pacificum*; RNA-seq libraries obtained from the siphonophores *Nanomia bijuga* and *Physalia*

*physalis*, *Craspedacusta sowerbyi*, *Ectopleura larynx*, *Podocoryna carnea* and two species of *Hydractinia, H. polyclina* and *H. symbiolongicarpus* (Table 1). For these specimens, we first captured several mitochondrial regions by mapping raw reads to the mtDNA from other hydrozoan genomes with Bowtie v.2 (*Langmead & Salzberg, 2012*) and MIRA v.4. We then extended these contigs with several rounds of baiting (using the mirabait script from MIRA v.4) and assembly (using the overlap-layout consensus assembler in Geneious) into gapped mtDNAs for *E. larynx*, *C. sowerbyi*, *L. tetraphylla*, *P. carnea*, *H. polyclina*, *H. symbiolongicarpus*, one specimen of *P. physalis* and *C. pacificum*, as well as the nearly-complete coding regions for *N. bijuga*, and another specimen of *P. physalis*.

We identified protein genes by blasting large (>300 bp) open reading frames (ORFs) obtained via translation using the minimally derived genetic code (translation table 4 = the Mold, Protozoan, and Coelenterate Mitochondrial Code) against published hydrozoan mtDNA genomes, followed by manual annotation. Transfer RNA (tRNA) genes were identified using the tRNAscan-SE and ARWEN programs (*Lowe & Eddy, 1997*; *Laslett & Canbäck, 2008*). We identified ribosomal (rRNA) genes by similarity (BLAST searches on NCBI's GenBank) to their counterparts in published mt-genomes and delimited the ends by alignment (see below).

## Sequence alignments and phylogenetic analyses

We prepared several multiple sequence alignments for phylogenetic analyses as described previously (*Kayal et al., 2013*). In short, the amino acid (AA hereafter) sequences of protein-coding genes were individually aligned using the L-INS-i option with default parameters of the MAFFT v.7 aligner online (*Katoh & Standley, 2013*) and subsequently concatenated. Nucleotide (NT hereafter) alignments for individual protein-coding genes were obtained according to their AA alignments using the online version of the PAL2NAL online program (*Suyama, Torrents & Bork, 2006*) and subsequently concatenated. Ribosomal genes (rRNA hereafter) were individually aligned using the online version of MAFFT with the Q-INS-i option (*Katoh & Toh, 2008*) and concatenated. We also created a concatenated all-nucleotides dataset consisting of NT and rRNA alignments (allNT hereafter). All concatenated alignments were filtered using Gblocks (*Talavera & Castresana, 2007*) with default parameters, allowing gaps in all positions, leading to alignments with 2,902 positions (2,501 informative sites) for AA, 9,864 positions (8,850 informative sites) for NT, 2,154 positions (1,664 informative sites) for rRNA, and 12,018 positions (10,773 informative sites) for allNT (Table S3, all alignments are provided as Supplemental Information). We estimated the number of phylogenetically informative sites with the DIVEIN online server (*Deng et al., 2010*), and the saturation levels of nucleotide alignments (NT, rRNA and allNT) using the DAMBE5 software (*Xia, 2013*).

We performed jModelTest v.2.1.4 (*Darriba et al., 2012*) and ProtTest v.3 (*Darriba et al., 2011*) on the nucleotide and amino acid alignments, respectively, to identify the most appropriate models of sequence evolution across entire alignments for subsequent phylogenetic analyses. Phylogenetic inferences were conducted under Maximum Likelihood framework using RAxML v.8 (*Stamatakis, 2014*) and under Bayesian framework using

MrBayes v.3.2.2 (*Ronquist et al., 2012*). Maximum Likelihood analyses were performed using the LG model of sequence evolution for amino acids. The General Time Reversible (GTR) models of nucleotide and amino acid evolution for all alignments were used for both Maximum Likelihood and Bayesian. Bayesian analyses consisted of two runs of 4 chains each of 10,000,000 generations using the GTR model for all alignments, sampled every 100 trees after a burn-in fraction of 0.25.

To investigate potential compositional biases in the datasets, amino acid and nucleotide composition of alignments were calculated using custom Python scripts (github.com/bastodian/shed/blob/master/Python/AA-Frequencies.py and github.com/bastodian/shed/blob/master/Python/GC-Frequencies.py), and visualized in 2-dimensional plots using the first two principal components as calculated by the princomp function in R version 2.15.1 (*R Core Team, 2014*).

## Evaluation of competing phylogenetic hypotheses

We tested 3 sets of traditional hypotheses of hydroidolinan relationships using likelihood-based topology tests with the approximately unbiased (AU) tests as implemented in Consel (*Shimodaira & Hasegawa, 2001*). Phylogenetic analyses were performed under the three following scenarios using constrained topological ML searches in PhyML v. 3.1 (*Guindon et al., 2010*) to calculate per-site likelihoods.

(1) Several studies have found Capitata to be the earliest branching clade within Hydroidolina e.g., (*Collins, 2002*; *Marques & Collins, 2004*; *Cartwright et al., 2008*; *Cartwright & Nawrocki, 2010*)) while another study suggested Aplanulata to be the earliest branch within Hydroidolina (*Collins et al., 2006*) . We compared these hypotheses to our best tree to evaluate if our data were able to reject either of these alternatives. (2) Filifera was traditionally viewed as being a monophyletic clade, but support for this nominal taxon has not been found so far (reviewed in *Collins, 2009*). We calculated the best ML tree under the constraint of the monophyly of Filifera and compared the resulting per-site likelihoods to those calculated from our best tree to evaluate if we can reject Filifera's monophyly given our datasets. (3) Anthoathecata (Aplanulata + Capitata) is a traditional taxon within Hydroidolina, a group not supported by our study and others (*Collins, 2009*); we compared the constrained topology containing monophyletic Anthoathecata to our best tree.

## RESULTS

### The mitochondrial genomes of hydrozoan cnidarians

We obtained partial or complete mtDNA from twenty-six hydrozoan species, more than tripling the number of mitogenomes available to date for this class. We found four different genome organizations in these hydrozoans (Fig. 1), three of which were described previously (*Kayal et al., 2012*): the trachymedusae *Geryonia proboscidalis* and *Liriope tetraphylla* have a mitochondrial genome organization similar to that known from other trachylines, *Cubaia aphrodite* (*Kayal et al., 2012*) and *Craspedacusta sowerbyi* (*Zou et al., 2012*); the mt genome organization in the aplanulatan *Euphysa aurata* is similar to those of other members of Aplanulata, *Ectopleura larynx* and *Hydra oligactis* (Fig. 1,

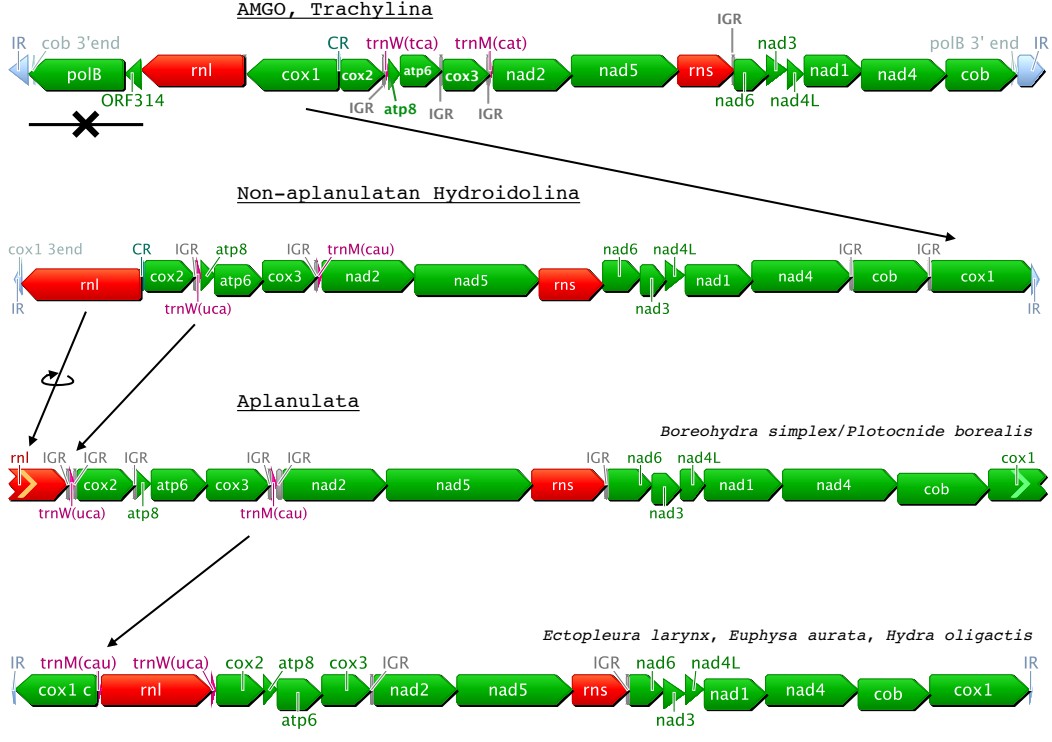

**Figure 1 Predicted evolution of the mitochondrial genome organization in Hydrozoa (Cnidaria).** Genes are color-coded as follows: green for proteins; red for rRNAs; purple for tRNAs; light-grey for repeated regions. CR: Control Region corresponding to the inversion of transcription orientation; IR: Inverted Repeat; IGR: Inter-Genic Region. AMGO corresponds to the Ancestral Mitochondrial Genome Organization as predicted in *Kayal et al. (2013)*; cox1 c is a duplicated *cox1* on the other end of the mtDNA; incomplete 5'end and 3'end are represented by chevrons on the left and right side of genes, respectively.

*Kayal et al., 2012*); the mt-genome organization in the species *Catablema vesicarium, Cladonema pacificum, Craseoa lathetica, Eudendrium capilare, Halitholus cirratus, Hydractinia polyclina, H. symbiolongicarpus, Leuckartiara octona, Melicertum octocostatum, Mitrocomella polydiademata, Nanomia bijuga, Podocoryna carnea, Ptychogena lactea, Rathkea octopunctata, Rhizophysa eysenhardti, Sarsia tubulosa*, and *Tiaropsis multicirrata*, as well as the partial mitogenome of *Proboscidactyla flavicirrata* are all similar to that of non-aplanulatan hydroidolinans described previously (*Kayal et al., 2012*). The mtDNA sequences of *Boreohydra simplex* and *Plotocnide borealis* were identical, confirming previous suggestions that these two names represent two stages in the life cycle of the same species (SV Pyataeva, RR Hopcroft, DJ Lindsay, AG Collins, 2015, unpublished data). Interestingly, the mt genome organization of this species is novel, potentially representing a transitional state between the mtDNA organization of other aplanulatan and that of non-aplanulatan hydroidolinans (Fig. 1, see 'Discussion').

We analyzed a large dataset of RNA-seq data (>230 M reads) from the siphonophore *Physalia physalis* and assembled the nearly-complete mt genome in multiple contigs. The mitochondrial genes represented only >7,000 reads (<0.003% of the total number

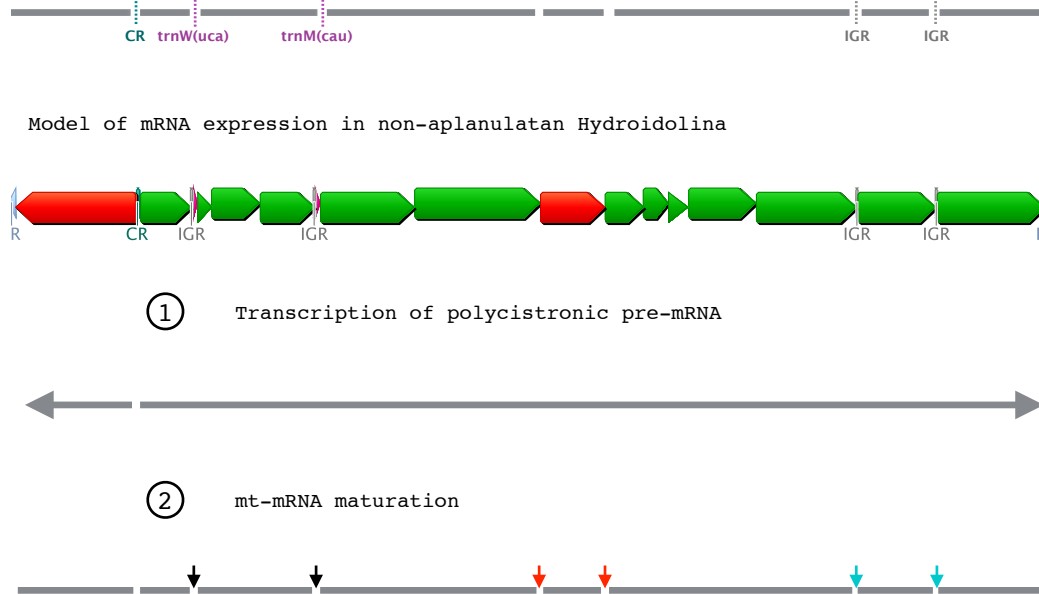

**Figure 2** **Mitochondrial gene expression in non-aplanulatan Hydroidolina.** (A) mtDNA organization in the siphonophore *Physalia physalis* assembled from a large EST dataset. Grey lines correspond to the contigs assembled. Missing features (tRNAs, IGRs, CR) are shown with dotted lines. (B) Predicted model of mt-mRNA expression based on findings from *P. physalis*. Color-codes are the same as Fig. 1. Grey horizontal arrows are the two pre-mRNA transcripts, the larger being polycistronic. Dark vertical arrows correspond to regions of pre-mRNA excision from the "tRNA punctuation model"; red and blue arrows are the additional excision sites predicted from our model for hydrozoan mt-mRNA expression. We predict that stage 1 and 2 are simultaneous.

of reads) of the *Physalia* RNA-seq data. We found both small and large ribosomal RNA subunits (*rns* and *rnl*, respectively) as well as the protein genes *cob*, *cox1*, and *cox2* in single-gene contigs. The other protein genes were found in collinear contigs as follows: *atp8-atp6-cox3*, *nad2-nad5*, and *nad6-nad3-nad4L-nad1-nad4* (Fig. 2A). We were not able to identify with enough confidence the two expected tRNA genes *mt-tRNA-Met* and *mt-tRNA-Thr* in this large RNA-seq dataset. Using an independently generated, smaller source of RNA-seq data (SRA Archive num. SRR871528), we assembled a more complete mt genome, confirming that the mtDNA organization in *Physalia physalis* was similar to that of the other siphonophore *Rhizophysa eysenhardti* obtained through long-range PCR. This smaller RNA-seq dataset provided the nearly-complete mtDNA sequence, with a few scattered gaps. The low amount of sequence data for *Nanomia bijuga* did not allow us to identify all the protein genes.

Genes were found to be very similar in length among all species (varying from identical to about 5% different in length). We found the GC content to be variable among the hydrozoan species we sampled, ranging 13.1–46.9% for protein coding genes, 19.4–39.1% for rRNA genes and 22.9–47.9% for tRNA genes. The most commonly used start codon

was ATG except for *atp8* in *Boreohydra simplex/Plotocnide borealis* (TTG) and *Tiaropsis multicirrata* (GTG); *cob* in *Tiaropsis multicirrata* (TTG); *cox1* in *Euphysa aurata* (GTG); *nad1* in *Physalia physalis* (GTG); *nad2* in *Catablema vesicarium*, *Halitholus cirratus*, and *Nanomia bijuga* (GTG); *nad3* in *Physalia physalis* (GTG); *nad4* in *Ptychogena lactea* (GTG); *nad4L* in *Nanomia bijuga* (GTG); *nad5* in *Boreohydra simplex/Plotocnide borealis* (TTG); *nad6* in *Catablema vesicarium* and *Nanomia bijuga* (GTG). TAA was the most commonly used stop codon for protein genes, with the exception of *nad5* and *nad6* where TAG was most often used (Table 2).

### Phylomitogenomics of Hydrozoa

Our AA, NT, rRNA, and allNT analyses under both Maximum Likelihood and Bayesian frameworks did not yield completely consistent results (Figs. 3 and S1–S8). Under GTR, the AA (Figs. S1 and S5), NT (Figs. S2 and S6) and allNT (Figs. 3 and S4) Maximum Likelihood and Bayesian analyses, respectively, yielded almost identical topologies, whereas the rRNA-based topologies (Figs. S3 and S7) and the AA topology assuming the LG model (Fig. S8) exhibited far lower resolution and support than all other topologies. The PCA of amino acid and nucleotide compositions (Fig. 4) of the alignments underlying our phylogenetic analyses, where taxa with similar composition cluster together, do not show evidence of strong compositional biases that may affect phylogenetic reconstruction.

Overall, we found a number of common relationships in all phylogenetic trees that were highly supported: the divergence between Trachylina and Hydroidolina within Hydrozoa, the monophyly of Leptothecata, Capitata *s.s.* and Aplanulata (Table 3). Within Hydroidolina, all analyses other than those based on just the rRNA data alone: (1) identified siphonophores as the first diverging clade in most trees; (2) supported Filifera I + Filifera II; (3) supported Aplanulata + Capitata s.s. + Filifera I–II; (4) supported Filifera III + Filifera IV, with the latter being paraphyletic with respect to the former in many trees, and; (5) supported Leptothecata + Filifera III–IV (Table 3). No analyses recovered Capitata in its former sense (Aplanaluta + Capitata *s.s.*) nor the monophyly of Anthoathecata or Filifera. Our constraint analyses show that the placement of Capitata or Aplanulata as the earliest branching clades within Hydroidolina is rejected by both NT and allNT (NT plus rRNA) alignments, whereas both AA and rRNA data alone cannot reject these hypotheses (Table S5). The monophyly of Filifera was rejected in all cases other than for the rRNA dataset (Table S5). Lastly, the monophyly of Anthoathecata was rejected for both NT and allNT datasets while AA and rRNA alignments do not reject this traditional hypothesis (Table S5).

## DISCUSSION

### The evolution of mtDNA in Hydrozoa

The gene arrangements of the newly sequenced hydrozoan mtDNAs are consistent with the three organizations recovered earlier (*Kayal et al., 2012*). For instance, the new trachyline mtDNAs exhibit the predicted organization of the ancestral mt-genome organization for Hydrozoa, with genes ordered into a small cluster (four genes, including the two

Kayal et al. (2015), *PeerJ*, DOI 10.7717/peerj.1403

**Table 2 Size, GC content and start and end codons for the genes of the newly obtained mtDNA.**

| Gene | | B.s. | C.v. | C.p. | C.s. RNA | E.l. RNA | E.c. | E.a. | G.p. | H.c. | H.p. | H.s. | L.o. | L.t. | M.o. | M.p. | N.b. | P.p. Y | P.p. SR | P.b. | P.c. | P.f. | P.l. | R.o. | R.e. | S.t. | T.m. |
|---|---|---|---|---|---|---|---|---|---|---|---|---|---|---|---|---|---|---|---|---|---|---|---|---|---|---|---|
| atp6 | size (nt) | 705 | 705 | 705 | 705 | 705 | 705 | 705 | 705 | 705 | 705 | 705 | 705 | 705 | 705 | 705 | 693 | 705 | 705 | 705 | 705 | ? | 705 | 705 | 705 | 705 | 705 |
| | GC | 22.4 | 26.4 | 24.8 | 45.1 | 28.3 | 22.6 | 22 | 30.1 | 26.2 | 25.2 | 26.4 | 24.4 | 29.8 | 27.1 | 29.4 | 25.7 | 32.3 | 32.9 | 22.3 | 25.7 | ? | 25.5 | 28.8 | 28.5 | 25.8 | 27.4 |
| | Start/End | A/A | A/A | A/A | A/A | A/A | A/A | A/A | A/A | A/A | A/A | A/A | A/A | A/A | A/A | A/A | A/A | A/A | A/A | A/A | A/A | ? | A/A | A/A | A/A | A/A | A/A |
| atp8 | size (nt) | 198 | 204 | 207 | 207 | 207 | 204 | 213 | 207 | 204 | 204 | 204 | 204 | 207 | 204 | 204 | ? | >126 | 204 | 198 | 204 | ? | 204 | 204 | 201 | 204 | 198 |
| | GC | 13.1 | 22.1 | 16.4 | 39.1 | 21.4 | 15.7 | 14.1 | 29.5 | 18.1 | 17.5 | 17.6 | 19.6 | 30 | 22.1 | 20.6 | ? | 20.6 | 24 | 13.1 | 21.6 | ? | 22.1 | 24.5 | 18.9 | 20.6 | 21.7 |
| | Start/End | T/A | A/A | A/A | A/A | A/A | A/A | A/A | A/A | A/A | A/A | A/A | A/A | A/A | A/A | A/A | ? | ?/A | A/A | T/A | A/A | ? | A/A | A/A | A/A | A/A | G/A |
| cob | size (nt) | 1,149 | 1,140 | 1,140 | 1,185 | >1024 | 1,143 | 1,140 | >787 | 1,140 | 1,143 | 1,143 | 1,140 | 1,203 | 1,143 | 1,146 | 1,164 | >1,016 | >1,133 | 1,149 | 1,143 | >848 | 1,146 | 1,146 | 1,143 | 1,143 | 1,146 |
| | GC | 25.5 | 27.5 | 27.8 | 46 | 29.7 | 25.9 | 26.3 | 36.5 | 27.3 | 27.5 | 27.8 | 27.6 | 32.3 | 29.3 | 31.8 | 29.2 | 36.5 | 35 | 25.4 | 29.1 | 25.9 | 27.4 | 33 | 28.4 | 28.6 | 30.6 |
| | Start/End | A/A | A/A | A/A | A/G | A/? | A/A | A/A | A/? | A/A | A/A | A/A | A/G | A/G | A/G | A/A | A/A | ?/? | ?/A | A/A | A/A | A/? | A/A | A/A | A/A | A/A | G/A |
| cox1 | size (nt) | >712 | >708 | 1,566 | 1,566 | >1,569 | >884 | >713 | >1,322 | >910 | 1,566 | 1,566 | >919 | 1,566 | >713 | >699 | 1,572 | >1,548 | 1,566 | >715 | 1,566 | >822 | >713 | >710 | >711 | >713 | >713 |
| | GC | 33 | 35.3 | 33.1 | 46.2 | 34.7 | 32.5 | 32 | 38 | 35.1 | 32.2 | 32.8 | 34.1 | 34.5 | 35.2 | 38.2 | 32.6 | 38.4 | 38.1 | 33 | 34.4 | 32.2 | 34.5 | 37.5 | 33.8 | 33.5 | 34.2 |
| | Start/End | A/? | A/? | A/G | A/G | ?/? | A/? | G/? | A/? | A/? | A/G | A/G | A/? | A/G | A/? | A/? | A/A | ?/A | A/A | A/? | A/G | ?/? | A/? | A/? | A/? | A/? | A/? |
| cox2 | size (nt) | 726 | 738 | 738 | 738 | >939 | 738 | 768 | 741 | 738 | 738 | 738 | 738 | 741 | 738 | 738 | >720 | >720 | 735 | 726 | 738 | ? | 738 | 738 | 735 | 738 | 735 |
| | GC | 25.8 | 29.5 | 28.6 | 44.3 | 27.6 | 28.6 | 22.3 | 37.2 | 29.7 | 28.1 | 28.2 | 29 | 32.1 | 30.8 | 34.1 | 31.4 | 39.2 | 38.5 | 25.8 | 29.7 | ? | 28.9 | 32 | 32.4 | 29.3 | 29.3 |
| | Start/End | A/A | A/A | A/A | ?/A | A/A | A/A | A/A | A/G | A/A | A/A | A/A | A/A | A/G | A/A | A/A | ?/G | ?/? | A/A | A/A | A/A | ? | A/A | A/A | A/A | A/A | A/A |
| cox3 | size (nt) | 786 | 786 | 786 | 786 | >778 | 786 | 786 | 786 | 786 | 786 | 786 | 786 | 786 | 786 | 786 | >798 | 786 | 786 | 786 | 786 | ? | 786 | 786 | 789 | 786 | 786 |
| | GC | 26.5 | 29.8 | 30.4 | 45.2 | 30.6 | 28.4 | 24 | 39.6 | 31.7 | 28.5 | 29.1 | 29.5 | 34.6 | 31.6 | 36.5 | 30.3 | 37.8 | 37.5 | 26.3 | 30 | ? | 31.7 | 32.8 | 32.6 | 32.1 | 33.3 |
| | Start/End | A/A | A/A | A/A | A/A | A/? | A/A | A/A | A/A | A/A | A/A | A/A | A/G | A/A | A/A | A/A | A/? | A/A | A/A | A/A | A/A | ? | A/A | A/A | A/A | A/A | A/G |
| nad1 | size (nt) | 978 | 990 | 987 | 999 | 981 | 990 | 990 | 999 | 990 | >972 | 990 | 990 | 999 | 987 | 987 | 990 | 990 | 990 | 978 | 990 | 987 | 990 | 987 | 987 | 990 | 990 |
| | GC | 24.7 | 28 | 26.6 | 46.9 | 28.7 | 27.2 | 23.6 | 37 | 30.5 | 25.2 | 26.7 | 27 | 33.4 | 28.5 | 31.9 | 26.2 | 36.1 | 35.8 | 24.7 | 27.8 | 24 | 27 | 30.1 | 30.4 | 27.5 | 28.5 |
| | Start/End | A/A | A/A | A/A | A/A | A/G | A/A | A/A | A/A | A/A | A/? | A/G | A/A | A/A | A/A | A/A | A/A | G/A | G/A | A/A | A/A | A/A | A/A | A/A | A/A | A/G | A/A |
| nad2 | size (nt) | 1,311 | 1,362 | 1,314 | 1,350 | >893 | 1,353 | 1,311 | 1,353 | 1,362 | >1,038 | 1,362 | 1,362 | 1,353 | 1,350 | 1,356 | ? | >1,176 | 1,338 | 1,311 | 1,356 | ? | 1,356 | 1,344 | 1,353 | 1,356 | 1,347 |
| | GC | 17.7 | 22.5 | 19.9 | 44 | 25.1 | 17.4 | 15.7 | 32.8 | 22.2 | 21 | 19.8 | 20.7 | 28.5 | 24.4 | 27 | ? | 32.2 | 31.3 | 17.8 | 21.2 | ? | 21.1 | 26.6 | 23.4 | 23.5 | 23.2 |
| | Start/End | A/A | G/A | A/A | A/A | ?/? | A/A | A/A | A/A | G/A | ?/? | A/A | A/A | A/A | A/A | A/A | ? | ?/A | A/A | A/A | A/A | ? | A/A | A/A | A/A | A/A | A/A |
| nad3 | size (nt) | 360 | 357 | 357 | 357 | 357 | 357 | 357 | 357 | 357 | 357 | 357 | 357 | 357 | 357 | 357 | 357 | 357 | 357 | 360 | 357 | 357 | 357 | 357 | 357 | 357 | 357 |
| | GC | 23.3 | 24.6 | 22.1 | 42.3 | 24.3 | 22.1 | 21.8 | 34.2 | 25.5 | 22.4 | 24.1 | 22.1 | 30.8 | 26.6 | 29.1 | 23.5 | 30 | 30.3 | 23.6 | 25.8 | 23.5 | 24.1 | 31.7 | 26.9 | 28 | 25.8 |
| | Start/End | A/A | A/A | A/A | A/A | A/A | A/A | A/A | A/A | A/A | A/A | A/A | A/A | A/A | A/A | A/A | A/A | G/A | G/A | A/A | A/A | A/A | A/A | A/A | A/A | A/A | A/A |
| nad4 | size (nt) | 1,455 | 1,458 | 1,458 | 1,461 | 1,458 | 1,449 | 1,458 | 1,461 | 1,458 | >1,251 | 1,458 | 1,458 | 1,461 | 1,455 | 1,455 | >974 | >1,111 | >1,315 | 1,455 | 1,458 | 1,446 | 1,455 | 1,458 | 1,461 | 1,458 | 1,455 |
| | GC | 22.6 | 25 | 24.3 | 44.6 | 26.3 | 21.5 | 20.2 | 35.8 | 24.6 | 24.7 | 24.1 | 23.6 | 32.2 | 26.7 | 29.9 | 27.3 | 33.1 | 33.1 | 22.7 | 25.7 | 23 | 25.5 | 29.4 | 26.8 | 26.1 | 27.2 |
| | Start/End | A/A | A/A | A/A | A/A | A/A | A/A | A/A | A/G | A/A | ?/A | A/A | A/A | A/G | A/A | A/G | ?/? | A/? | A/? | A/A | A/A | A/A | G/A | A/A | A/G | A/A | A/A |

Table 2 (*continued*)

| Gene | | B.s. | C.v. | C.p. | C.s. RNA | E.l. RNA | E.c. | E.a. | G.p. | H.c. | H.p. | H.s. | L.o. | L.t. | M.o. | M.p. | N.b. | P.p. Y | P.p. SR | P.b. | P.c. | P.f. | P.l. | R.o. | R.e. | S.t. | T.m. |
|---|---|---|---|---|---|---|---|---|---|---|---|---|---|---|---|---|---|---|---|---|---|---|---|---|---|---|---|
| *nad4L* | size (nt) | 297 | 300 | 294 | 300 | 300 | 300 | 300 | 300 | 300 | 300 | 300 | 300 | 300 | 300 | 300 | >279 | 297 | 297 | 297 | 300 | 297 | 300 | 297 | 300 | 297 | 300 |
| | GC | 20.5 | 20.7 | 21.1 | 40 | 21.7 | 20.7 | 19.7 | 29.3 | 21.3 | 16.9 | 19 | 20.3 | 26.3 | 24 | 24.3 | 26.9 | 26.9 | 27.3 | 20.5 | 20.3 | 19.9 | 22 | 25.3 | 22.7 | 23.9 | 26.3 |
| | Start/End | A/A | A/A | G/A | A/A | A/A | A/A | A/A | A/A | A/A | A/A | A/A | A/A | A/A | A/A | A/A | G/G | A/G | A/G | A/A | A/A | A/A | A/A | A/A | A/A | A/A | A/A |
| *nad5* | size (nt) | 1,827 | 1,833 | 1,830 | 1,833 | >1,730 | 1,830 | 1,853 | 1,833 | 1,833 | >1,672 | 1,833 | 1,833 | 1,833 | 1,833 | 1,833 | >1,813 | >1,719 | 1,830 | 1,830 | 1,833 | >1,137 | 1,833 | 1,833 | 1,830 | 1,833 | 1,833 |
| | GC | 21.1 | 25.3 | 25.1 | 46.3 | 28.1 | 22.8 | 20.5 | 36.1 | 26.6 | 24.7 | 23.5 | 24.4 | 32.3 | 26 | 29.6 | 24.8 | 34.1 | 34.1 | 21 | 25.6 | 22.1 | 25 | 27.9 | 27.1 | 26.7 | 27.1 |
| | Start/End | T/A | A/G | A/G | A/A | ?/? | A/G | A/A | A/A | A/G | ?/? | A/G | A/G | A/A | A/G | A/G | ?/? | A/? | A/A | T/A | A/G | A/A | A/G | A/G | A/A | A/G | A/G |
| *nad6* | size (nt) | 567 | 561 | 549 | 564 | 558 | 552 | 555 | 564 | 561 | 564 | 564 | 561 | 564 | 564 | 564 | 552 | >555 | 561 | 567 | 564 | 564 | 564 | 558 | 561 | 552 | 561 |
| | GC | 19.2 | 23 | 21.3 | 44 | 22.4 | 18.8 | 17.5 | 30.3 | 24.1 | 20.2 | 20.6 | 20.9 | 28 | 25.4 | 25.4 | 20.1 | 32.4 | 33 | 19.2 | 23.8 | 19.1 | 21.6 | 28.5 | 26.6 | 23.7 | 25 |
| | Start/End | A/A | G/G | A/G | A/G | A/A | A/G | A/A | A/G | A/G | A/G | A/G | A/G | A/G | A/G | A/G | ?/G | ?/G | A/G | A/A | A/G | A/A | A/G | A/G | A/G | A/G | A/A |
| *rnl* | size (nt) | >743 | >1,592 | 1,720 | >1,784 | >1,784 | >1,303 | >775 | >600 | >1,590 | 1,733 | 1,737 | >892 | 1,768 | >1,591 | >1,581 | >1,759 | 1,761 | 1,756 | >742 | 1,704 | >505 | >1,58 | >1,58 | >1,61 | >1,58 | >1,583 |
| | GC | 24.9 | 23.9 | 23.3 | 36.5 | 26.4 | 20.3 | 23.7 | 39.1 | 24.5 | 23.9 | 24.8 | 19.4 | 27.1 | 24.5 | 27.2 | 22 | 32.3 | 32.5 | 24.9 | 24.1 | 26.9 | 24.4 | 31.7 | 23.5 | 24.4 | 26.3 |
| *rns* | size (nt) | 930 | 930 | 912 | 995 | >865 | 925 | 896 | 969 | 925 | >910 | 931 | 928 | 968 | 921 | 918 | >845 | >873 | 922 | 930 | 931 | 906 | 920 | 902 | 926 | 915 | 920 |
| | GC | 21 | 27.8 | 23.6 | 37.6 | 24.9 | 25.3 | 22.5 | 31.3 | 28.1 | 25.6 | 25.6 | 27.6 | 29.1 | 25.7 | 26.6 | 23.2 | 32 | 31.5 | 21 | 24.9 | 25.1 | 26.5 | 32.9 | 26.6 | 25 | 29.7 |
| *trnM* | size (nt) | 74 | 71 | 71 | 71 | ? | 71 | ? | 71 | 71 | ? | 71 | 71 | 71 | 71 | 69 | ? | ? | 69 | 74 | 67 | ? | 69 | 69 | 69 | 71 | 69 |
| | GC | 24.3 | 23.9 | 33.8 | 36.6 | ? | 26.8 | ? | 28.2 | 23.9 | ? | 29.6 | 23.9 | 26.8 | 26.8 | 30.4 | ? | ? | 37.7 | 24.3 | 28.4 | ? | 27.5 | 34.8 | 27.5 | 31 | 30.4 |
| *trnW* | size (nt) | 70 | 70 | 70 | 71 | ? | 70 | 70 | 71 | 70 | ? | 70 | 70 | 71 | 70 | 70 | ? | ? | ? | 70 | 70 | ? | 70 | 70 | 70 | 70 | 70 |
| | GC | 25.7 | 35.7 | 34.3 | 47.9 | ? | 32.9 | 30 | 40.8 | 35.7 | ? | 34.3 | 41.4 | 39.4 | 35.7 | 32.9 | ? | ? | ? | 25.7 | 34.4 | ? | 28.6 | 44.3 | 28.6 | 34.3 | 35.7 |
| *orf314* | size (nt) | NA | NA | NA | 354 | NA | NA | NA | ? | NA | NA | NA | NA | 297 | NA | NA | NA | NA | NA | NA | NA | NA | NA | NA | NA | NA | NA |
| | GC | NA | NA | NA | 37.9 | NA | NA | NA | ? | NA | NA | NA | NA | 22.2 | NA | NA | NA | NA | NA | NA | NA | NA | NA | NA | NA | NA | NA |
| | Start/End | NA | NA | NA | A/A | NA | NA | NA | ? | NA | NA | NA | NA | A/A | NA | NA | NA | NA | NA | NA | NA | NA | NA | NA | NA | NA | NA |
| *polB* | size (nt) | NA | NA | NA | 1,668 | NA | NA | NA | ? | NA | NA | NA | NA | 1,644 | NA | NA | NA | NA | NA | NA | NA | NA | NA | NA | NA | NA | NA |
| | GC | NA | NA | NA | 41.7 | NA | NA | NA | ? | NA | NA | NA | NA | 30.4 | NA | NA | NA | NA | NA | NA | NA | NA | NA | NA | NA | NA | NA |
| | Start/End | NA | NA | NA | A/A | NA | NA | NA | ? | NA | NA | NA | NA | A/A | NA | NA | NA | NA | NA | NA | NA | NA | NA | NA | NA | NA | NA |

**Notes.**

B. s., *Boreohydra simplex*; C. v., *Catablema vesicarium*; C. p., *Cladonema pacificum*; C. s. RNA, *Craspedacusta sowerbyi*; E.l. RNA, *Ectopleura larynx*; E. c., *Eudendrium capillare*; E. a., *Euphysa aurata*; G. p., *Geryonia proboscidalis*; H. c., *Halitholus cirratus*; H. p., *Hydractinia polyclina*; H. s., *Hydractinia symbiolongicarpus*; L. o., *Leuckartiara octona*; L. t., *Liriope tetraphylla*; M. o., *Melicertum octocostatum*; M. p., *Mitrocomella polydiademata*; N. b., *Nanomia bijuga*; P. p. Y, *Physalia physalis Y*; P. p. SR, *Physalia physalis SR*; P. b., *Plotocnide borealis*; P. c., *Podocoryna carnea*; P. f., *Proboscidactyla flavicirrata*; P. l., *Ptychogena lactea*; R. o., *Rathkea octopunctata*; R. e., *Rhizophysa eysenhardti*; S. t., *Sarsia tubulosa*; T. m., *Tiaropsis multicirrata*.

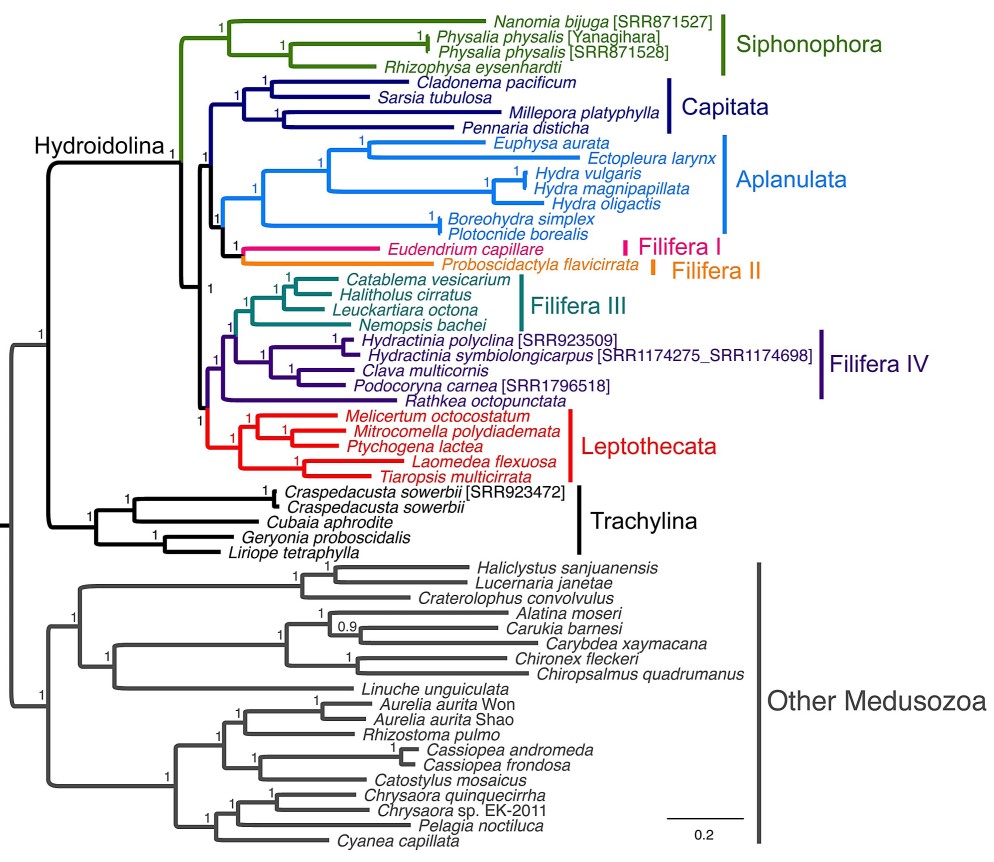

**Figure 3** **Phylogenetic analysis of the allNT alignment under the Bayesian framework using MrBayes with the GTR+Γ model of sequence evolution.** Support values correspond to posterior probabilities.

extra protein genes *polB* and *orf314*) and a large cluster (thirteen genes) with opposite orientations (*Kayal et al., 2012*). We note, however, that our taxon sampling within Trachylina is still relatively limited, restricted to representatives of Limnomedusae plus the Trachymedusae *Liriope tetraphylla* and *Geryonia proboscidalis*, which have been shown to be more closely related to Limnomedusae than to other members of Trachymedusae (*Collins et al., 2008*). Thus, the possibility remains that other trachyline taxa (including Narcomedusae, Actinulida and other members of Trachymedusae) could exhibit an as yet unidentified mt-genome organization. New taxon sampling within Hydroidolina shows that hydroidolinan mtDNA organization is nearly identical to that so far observed in trachylines, except that they lack, and likely lost (*Kayal et al., 2012*), the two non-standard protein-coding genes *polB* and *orf314*. Gene organizations within Aplanulata are the most derived from the putative ancestral one for Hydroidolina, where all genes are in the same orientation but the second copy of *cox1* (which can be partial) oriented in the opposite direction to the rest of the genome (Fig. 1). Our new data are partially consistent with the proposed scenario for the evolution of the mitochondrial genome organization in Hydrozoa (*Kayal et al., 2012*). Specifically, the ancestral hydrozoan mtDNA contained the two extra protein-coding genes *orf314* and *polB*, which were subsequently

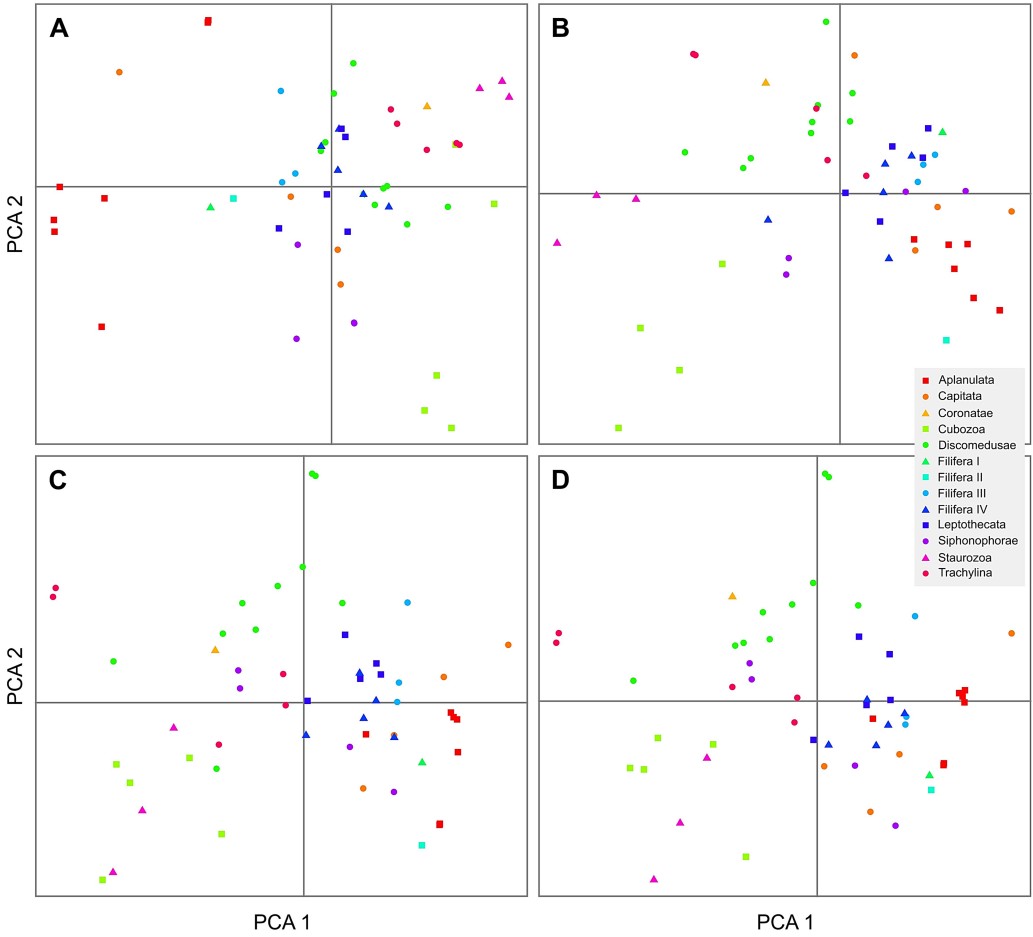

**Figure 4** 2-dimensional plots of the first two principal components from the principal component analysis of the composition of the AA (A), rRNA (B), NT (C) and allNT (D) alignments.

lost in Hydroidolina before the divergence of various orders. Aplanulata displays not two, but three increasingly derived genome organizations generated by sequential gene rearrangement (Fig. 1): inversion of *rnl* and translocation of *trnW* in *Boreohydra simplex/Plotocnide borealis*; translocation of *trnM* in *Ectopleura larynx*, *Euphysa aurata* and *Hydra oligactis*; partitioning of the genome into two nearly equal-sized chromosomes in some species of *Hydra* (*Kayal et al., 2012*). We found inter-genic regions (IGRs) longer than 10 bp after *cox2* in *Boreohydra simplex/Plotocnide borealis* and after *cox3* in *Ectopleura larynx*, *Euphysa aurata* and *Hydra* spp. These IGRs could conceivably be residues left from the translocations of *trnW* and *trnM*, respectively, but no obvious homology was found in our alignments (data not shown).

## Expression of mtDNA genes in hydrozoans

In Metazoa, mitochondrial gene expression is thought to follow the "tRNA punctuation model," where mt genes are transcribed into polycistronic precursor transcripts (*Ojala, Montoya & Attardi, 1981*; *Gissi & Pesole, 2003*), followed by the excision of the tRNAs that

Table 3 Posterior probabilities and bootstrap values for different clades within Hydrozoa.

| | MB | | | | ML | | | | |
|---|---|---|---|---|---|---|---|---|---|
| | aa(GTR) | NT(GTR) | rRNA(GTR) | allNT(GTR) | AA(GTR) | AA(LG) | NT(GTR) | rRNA(GTR) | allNT(GTR) |
| Aplanulata | 1 | 1 | 1 | 1 | 100 | 100 | 100 | 88 | 100 |
| Capitata | 0.95 | 1 | 1 | 1 | NA | NA | 63 | 94 | 99 |
| Filifera | NA | NA | NA | NA | NA | NA | NA | NA | NA |
| Leptothecata | 1 | 1 | 1 | 1 | 100 | 100 | 100 | 85 | 100 |
| Anthoathecata | NA | NA | NA | NA | NA | NA | NA | NA | NA |
| Filifera I + II | 0.91 | 1 | NA | 1 | 41 | 42 | 75 | NA | 80 |
| Apla + Capit | NA | NA | NA | NA | NA | NA | NA | NA | NA |
| Apla + Fili I–II | 0.61 | 1 | NA | 1 | NA | NA | 36 | NA | 66 |
| Apla + Capit + Fili I–II | NA | 1 | NA | 1 | NA | NA | 80 | NA | 81 |
| Lepto + Fili III–IV | 1 | 1 | NA | 1 | 91 | 91 | 91 | NA | 91 |
| Sipho + Apla | NA | NA | NA | NA | NA | NA | NA | NA | NA |
| Sipho + Lepto | NA | NA | NA | NA | NA | NA | NA | NA | NA |
| Antho + Lepto | 0.68 | 1 | NA | 1 | NA | NA | 78 | NA | 80 |

release single-gene (monocistronic) mRNAs and rRNAs (*Mercer et al., 2011*). Unlike most animals, cnidarian mtDNAs encode either one (*trnM* in Cubozoa and Octocorallia) or two (*trnM* and *trnW* in the remaining taxa) tRNA genes. This begs the question of the mechanisms involved in the expression of mt genes for this group.

RNA-seq studies provide unique insights into the expression of genes, and we used data obtained through RNA-sequencing projects to better understand translational mechanisms of the linear mtDNAs in hydrozoans. Surveying several large RNA-seq datasets on NCBI's GenBank and one from an unpublished source (Table 1), we assembled and annotated the nearly complete mtDNA sequences for eight hydroidolinan Hydrozoa species, including three of the first four mtDNA genomes from representatives of Siphonophorae. For all non-aplanulatan hydroidoline hydrozoans species, we found no RNA-seq reads upstream of *cox2* and *rnl*, a large intergenic region (IGR) that marks the inversion of the transcriptional orientation of mitochondrial genes (Fig. 1). It was previously suggested that this IGR has the potential to fold into a stem-loop, serving as the putative mt control region (CR) in non-aplanulatan hydroidoline hydrozoans (*Kayal et al., 2012*); our results further support this hypothesis. In fact, two large RNA-seq runs from the filiferan *Hydractinia symbiolongicarpus* (SRA Archive num. SRR1174275 and SRR1174698) and one from *Podocoryna carnea* (SRA Archive num. SRR1796518) allowed assembling the complete mtDNA for these species excluding the CR, with an organization similar to that of other non-aplanulatan hydrozoans. This pattern suggests that the mtDNA is transcribed into two polycistronic precursor transcripts (mt pre-mRNA) with opposite orientations (Fig. 2B). Surprisingly, the CR of *Craspedacusta sowerbyi* (SRA Archive num. SRR923472) was mapped onto a few RNA-seq reads. We believe that this particular dataset contains some DNA sequences, perhaps resulting from contamination of the original cDNA libraries by mitochondrial DNA.

In siphonophores, as in other non-aplanulatan hydroidolinans, the *trnW* gene is situated between *cox2* and *atp8*, while the *trnM* gene falls between *cox3* and *nad2* (Fig. 1). While we expect the mt genome of the siphonophore *Physalia physalis* to be organized into a single chromosome similar to that of *Rhizophysa eysenhardti* as suggested by the small RNA-seq data, the partial mt genome obtained from the large RNA-seq data assembled into eight contigs (Fig. 2A). The smaller RNA-seq dataset produced six contigs, including polycistronic *trnW*(3'end)-*atp8-atp6-cox3-trnM-nad2*(5'end), *nad2*(partial)-*nad5*(partial), *nad5*(3'end)-*rns-nad6-nad3-nad4L-nad1-nad4*(partial) and *cob*(partial)-*cox1*, as well as monocystronic *cox2* and *rnl*. The failure to recover full-length genes likely resulted from insufficient coverage of mt-RNAs in this dataset. For the larger RNA-seq dataset, we found a different pattern of gaps, none within genes; reads span across protein gene boundaries for *atp8-atp6-cox3*, *nad2-nad5*, and *nad6-nad3-nad4L-nad1-nad4* (Fig. 2A) with average coverage ranging from 8 to 223 reads per contig (data not shown). The absence of any reads between these gene clusters, as well as between contigs *nad2-nad5* and *rns* or *rns* and *nad6-nad3-nad4L-nad1-nad4* in the large RNA-seq data neither appears to be the result of insufficient read depth nor is it easily explained by the highly transient nature of the polycistronic precursor (pre-mRNA) transcript. The two sets of *Physalia* RNA data were produced using different approaches for capturing mRNAs and building the Illumina libraries, resulting in different maturation levels of the transcripts. We posit that the larger RNA-seq dataset contains only mature mt-mRNAs while the smaller RNA-seq dataset has both pre- and mature mRNAs. Accordingly, the pattern of mt-RNA expression is in part in accord with the tRNA punctuation model, where the excision of the tRNAs would release monocistronic *cox2* and polycistronic *atp8-atp6-cox3* from the pre-mRNA (Fig. 2B, Step 2, black arrows). Yet, this model does not explain the bicistronic *nad2-nad5* nor monocistronic *rns*, *cob* and *cox1*. It is possible that both the rRNAs and the tRNAs are excised, simultaneously or sequentially, from the precursor transcript, releasing bicistronic *nad2-nad5* and monocistronic *rns* (Fig. 2B, Step 2, red arrows). However, an additional mechanism would need to be invoked to explain the excision of *cob* and *cox1* (as illustrated by the absence of reads spanning across that gene boundary) from the polycistronic precursor transcripts. We observed intergenic regions of 10 bp or longer with conserved motifs in these positions (Figs. 2 and S9) with potential secondary structures (Figs. S10) that could represent recognition sites for the enzyme involved in maturation of mRNA (Fig. 2B, Step 2, blue arrows). This scenario is supported by the presence of IGRs before (and sometimes after) mt-tRNAs. In fact, by forming short stem-loops, these IGRs might signal for the maturation of mt pre-mRNA in Hydrozoa in a similar fashion as mt-tRNAs in other animals (*Mercer et al., 2011*).

## Mitochondrial view of hydrozoan character evolutionary history

Using the coding regions of the mtDNA from thirty-seven hydrozoan species, including twenty-six newly obtained for this study, we inferred the evolutionary history of Hydrozoa. To date, most studies of hydrozoan phylogeny have relied on rRNA sequence data, providing some important insights, but no reliable inferences of relationships
among hydroidolinan taxa (*Collins et al., 2006*; *Cartwright et al., 2008*; *Cartwright & Nawrocki, 2010*). In our analyses, we similarly found mt-rRNA insufficient for deciphering relationships among hydroidolinan lower clades with high support (Figs. S3, S7 and Table 3). The saturation test (*Xia et al., 2003*) suggests a high level of saturation in the rRNA alignment for 16 and 32 OTUs, while saturation levels are assumed acceptable for the other datasets (Table S2), which could explain the poor performance of rRNA. Similarly, rRNA alone did not allow discriminating among several competing hypotheses of hydroidolinan relationships while NT data did (Table S5).

Our phylogenetic analyses strongly support the monophyly of Trachylina and Hydroidolina, while rejecting Anthoathecata and Filifera as suggested by other molecular data (*Cartwright et al., 2008*; *Cartwright & Nawrocki, 2010*). Interestingly, our data support the hypothesis that Siphonophorae is the first diverging lineage within Hydroidolina (Fig. 3 and Table 3) in contrast to a recent phylogenomic study that found Aplanulata to be the earliest branching clade within Hydroidolina while Siphonophorae was nested within Hydroidolina (*Zapata et al., 2015*). Previous studies have grouped, though with low support, Siphonophorae with either Aplanulata (*Cartwright et al., 2008*) or Leptothecata (*Cartwright et al., 2008*; *Cartwright & Nawrocki, 2010*), but both hypothetical positions are contradicted by our analyses (Table 3). Our competing hypothesis suggests that the unique holopelagic colonial organization of siphonophores could have been an early innovation within Hydrozoa. However, given that it is apomorphic, it could have evolved anywhere along the lineage leading from the origin of Hydroidolina to the last common ancestor of Siphonophorae.

Recent rRNA phylogenetic studies have broken Filifera into four clades (I–IV), with varying levels of support (*Cartwright et al., 2008*). As with our data, rRNA data revealed a clade, albeit with low support, uniting Filifera I (= family Eudendriidae), Filifera II, and Aplanulata. Similarly, our results are consistent with the rRNA-based results, again with low support, that Filifera III and Filifera IV form a clade. However, mitochondrial genome data suggest that Filifera III is embedded within Filifera IV. Studies on morphology and rRNA data have placed *Clava multicornis* within Hydractiniidae, making it a member of Filifera III (*Schuchert, 2001*; *Cartwright et al., 2008*), which is confirmed by our results. Interestingly, in our trees Filifera IV was found to include a poorly supported, but morphologically distinct, clade dubbed Gonoproxima, containing species that do not bear gonophores on the hydranth body, instead budding on the hydrocauli, pedicels, or stolons (*Cartwright et al., 2008*; *Cartwright & Nawrocki, 2010*). Our taxon sampling is much more depauperate, but our analyses suggest that the positioning of the gonophores may perhaps be evolutionarily too labile to be strictly used for classification, similar to the presence of scattered tentacles (*Schuchert, 2001*).

The well-supported clade formed by [Aplanulata + (Filifera I + II) + Capitata] is an interesting result, but our taxon sampling is too limited to make strong conclusions about whether the capitate tentacles of Aplanulata and Capitata are shared derived characters (with a reversal in the lineage leading to Filifera I + Filifera II), or whether they evolved

independently. It is not surprising that the absence of capitate tentacles (the main uniting feature of Filifera) is not revealed to be a synapomorphy.

Significantly more than half of the species within Hydroidolina are contained within Leptothecata, which highlights the lack of taxon sampling in our analysis with just five species represented. Ribosomal analyses have revealed *Melicertum octocostatum*, a species that actually lacks a theca in the hydroid stage, to be of the sister taxon to the remainder of Leptothecata (*Cartwright et al., 2008*; *Leclère et al., 2009*; *Cartwright & Nawrocki, 2010*), raising the possibility that the theca was derived within Leptothecata rather than emerging right at its base. Our analyses also contain *Melicertum octocostatum* diverging early within Leptothecata, but not sister to all other sampled leptothecates. Given the caveat that taxon sampling is limited, the absence of a theca in *Melicertum* is likely a secondary loss. In fact, several other leptothecates show a reduced or diminutive theca into which the hydranth is not able to retract.

## CONCLUSION

In this study, we assembled and annotated twenty-three novel nearly-complete or complete mitochondrial genomes from most orders of the class Hydrozoa, with an emphasis on the subclass Hydroidolina. Increased taxon sampling revealed only one additional mitogenome organization beyond those described previously for hydrozoans, being consistent with the most recent overall picture of mitogenome evolution (*Kayal et al., 2012*). Using EST data, we proposed that the mitochondrial pre-mRNA is polycistronic, with tRNAs and rRNAs likely excised simultaneously during transcription following a modified tRNA punctuation model. Using both nucleotide and amino acid alignments, we inferred the evolutionary history of taxa within Hydroidolina, one of the most difficult questions in cnidarian phylogenetics. In contrast to previous analyses, our data yield resolved topologies and provide a working hypothesis for deep hydroidolinan relationships. Specifically, mitogenome data suggest that Siphonophorae is the earliest diverging group within Hydroidolina; a clade is formed by Leptothecata + Filifera III/IV, where Filifera IV/Gonoproxima is paraphyletic; and Aplanulata/Capitata/Filifera I + II form a clade. We conclude that mitochondrial protein coding sequence data is a pertinent marker for resolving the phylogeny of Hydrozoa. Future investigations of hydrozoans could take advantage of the highly conserved mitogenome organization and the ever-decreasing price of sequencing to obtain the complete mtDNA for massive numbers of hydrozoan samples. We are looking forward to additional studies using alternate data (nuclear genes and genomes) to test our findings.

## ACKNOWLEDGEMENTS

We would like to thank Matthew Kwenskin for his help with installing and modifying the bioinformatics tools and Niamh Redmond for her assistance in developing the library preparation protocols. Much of this work was performed using resources of the Laboratories of Analytical Biology at the Smithsonian National Museum of Natural History. Peter Schuchert is gratefully acknowledged for providing specimens/DNA extracts and for providing feedback on an earlier version of this MS.

### Funding

This project was mainly funded by a Smithsonian Institution Peter Buck Predoctoral Grant to EK. This work was also supported by the National Science Foundation's Assembling the Tree of Life program (DEB No. 0829986 to R.W. Thacker and A.G.C.). BB wishes to acknowledge an NSF Doctoral Dissertation Improvement Grant (DEB 0910237) that funded the sequencing of *Liriope tetraphylla*. RRH was funded in part by the National Oceanic and Atmospheric Administration under UAF cooperative agreements NA13OAR4320056 and NA08OAR4320870, and by the project BOFYGO (from the board of the Danish Centre for Marine Research, DCH) and the Greenland Climate Research Centre (project 6505) via TGN. *Physalia physalis* sequencing was supported by NIH/NIAMS, R01 AR059388 to AY. The funders had no role in study design, data collection and analysis, decision to publish, or preparation of the manuscript.

### Grant Disclosures

The following grant information was disclosed by the authors:
Smithsonian Institution Peter Buck Predoctoral.
National Science Foundation's Assembling the Tree of Life program: 0829986.
NSF Doctoral Dissertation Improvement: 0910237.
National Oceanic and Atmospheric Administration under UAF: NA13OAR4320056, NA08OAR4320870.
Danish Centre for Marine Research, DCH.
Greenland Climate Research Centre: 6505.
NIH/NIAMS: R01 AR059388.

### Competing Interests

The authors declare there are no competing interests.

### Author Contributions

- Ehsan Kayal conceived and designed the experiments, performed the experiments, analyzed the data, contributed reagents/materials/analysis tools, wrote the paper, prepared figures and/or tables, reviewed drafts of the paper.
- Bastian Bentlage performed the experiments, analyzed the data, contributed reagents/materials/analysis tools, wrote the paper, prepared figures and/or tables, reviewed drafts of the paper.
- Paulyn Cartwright, Angel A. Yanagihara, Dhugal J. Lindsay and Russell R. Hopcroft contributed reagents/materials/analysis tools, reviewed drafts of the paper.
- Allen G. Collins conceived and designed the experiments, contributed reagents/materials/analysis tools, wrote the paper, prepared figures and/or tables, reviewed drafts of the paper.

## DNA Deposition

The following information was supplied regarding the deposition of DNA sequences:

A majority of the newly described mitochondrial genomes have being submitted to GenBank: KT809334, KT809330, KT809334, KT809323, KT809333, KT809324, KT809336, KT809337, KT809325, KT809319, KT809329, HT809320, KT809321, KT809332, KT809322, KT809326, KT809328, KT809335, KT809331, KT809327. The rest have been submitted to the European Nucleotide Archive (ENA): LN901194, LN901195, LN901196, LN901197, LN901198, LN901199, LN901200, LN901201, LN901202, LN901203, LN901204, LN901205, LN901206, LN901207, LN901208, LN901209, LN901210.

## Data Availability

GenBank: KT809334, KT809330, KT809334, KT809323, KT809333, KT809324, KT809336, KT809337, KT809325, KT809319, KT809329, HT809320, KT809321, KT809332, KT809322, KT809326, KT809328, KT809335, KT809331, KT809327. ENA: LN901194, LN901195, LN901196, LN901197, LN901198, LN901199, LN901200, LN901201, LN901202, LN901203, LN901204, LN901205, LN901206, LN901207, LN901208, LN901209, LN901210.

## Supplemental Information

Supplemental information for this article can be found online at http://dx.doi.org/10.7717/peerj.1403#supplemental-information.

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
