# Peer review of "Phylogenetic analysis of higher-level relationships within Hydroidolina (Cnidaria: Hydrozoa) using mitochondrial genome data and insight into their mitochondrial transcription"

_PeerJ, doi:10.7717/peerj.1403_

## Round 0.1 · original submission · Minor Revisions

Please incorporate the minor suggestions given by both reviewers, which will improve the final version of your manuscript. Congratulations.

Reviewer 1 ·

Basic reporting

In general the manuscript conforms to basic formatting expectations, but there are a few aspects of the presentation of the material that could be improved:

1. The Introduction would benefit from some revision: it is confusing in some places and simply not very informative in others. Specifically, the paragraph from 64-86 that provides an overview of the higher taxonomy of Hydroidolina is quite confusing. It's not clear if the clades that are mentioned have been recognized (and received support) based on morphology alone, molecular analyses, or some combination of the two. Siphonophorae is first stated to be a well-supported monophyletic clade (67), but several sentences later (74) it's listed as a clade within Anthoathecata. 78-79 states that Aplanulata is a well-supported clade but that it includes members of Capitata, which has also just been referred to as a clade. Clarifying what clades have been established based on morphological evidence, and have subsequently held up (or not) to molecular analysis might help make this paragraph easier to follow. Further down in the Introduction there are two paragraphs that don't really serve much obvious function. 96-103 states that we now have the ability to use whole mt genomes for phylogenetic inference but doesn't include any information about why this approach might be more useful for phylogenetic inference than single-gene (or nuclear gene) approaches... And 104-112 posits that there are challenges to sequencing linear genomes, but doesn't indicate specifically what those are (presumably getting the ends of the sequences by PCR?) or how they have been or can be overcome.

2. The Figure legends and Table titles could be more informative than they are. Some figure elements and abbreviations are not defined. Some examples:
- Fig. 2 refers to "light" arrows, but not to the red and blue arrows that are shown.
- No mention is made of the light green chevrons that are in Figs 1 and 2 - what do they signify?
- What is "cox1 c" in Fig 1D?
- The text references a clade called "Gonoproxima" which is not indicated in Fig. 3.
- Table 1 does not state where vouchers are housed, or what "Notes" means (the SRR numbers are referred to at several points in the text, but what do they signify)?
- In Table 3, Start/Stop codons are given as e.g. A/A. Does that mean ATG/ATA? What does "NA" mean (clade not recovered? clade not supported?)?

3. The Results section is uneven. Some results that should be described in the text are glossed over and the reader is left to figure things out from the figures, while other information that is adequately presented in Tables is unnecessarily repeated in the text. Paragraph 202-217 is just a list of which species had the same gene orders, but doesn't tell the reader what those gene orders are or how they differ from one another. The reader has to figure that out from Fig.1. And one genome is stated to be "novel" and "transitional" but what about it makes it those things is not described. Rather than simply listing what species have the same gene arrangements, it would be much more informative to describe how the arrangements differ from one another. A Table might be a more efficient way to then list which species have which arrangement. Conversely, the information given in paragraph 229-238 is all summarized in Table 2, and doesn't need to be repeated in the text (although see note above about defining abbreviations). It's adequate just to state that some genes used alternative start and stop codons, and refer the reader to the table for the specific information about which ones.

Experimental design

The methods used here to sequence and bioinformatically infer mt genomes were varied, but seem fairly straightforward. The Abstract, however, highlights the fact that "we describe a new, relatively cheap and accessible multiplexing strategy..." But this new method is described entirely within a Supplement to the manuscript. If this is a novel enough protocol to highlight in the Abstract, shouldn't it be described within the main body of the paper?

Validity of the findings

The findings fall into three separate categories: 1. Inference of mt genome maps. 2. Use of the nt and AA sequence data to infer phylogenetic relationships. 3. Speculation about the mechanism of mt gene transcription in Hydrozoa. I have no concerns about the analysis and interpretation of the phylogenetic results, but some additional information would help support the authors' conclusions regarding the genome maps and proposed model of transcription.

1. The text refers to the "difficulty" of sequencing linear genomes, and repeatedly states that "nearly complete" or "more-or-less complete" genomes were obtained. What does this mean? How complete or incomplete were the genomes? (And what is the distinction between "near-complete" and "more-or-less complete"? - lines 163-164 imply one has been made.) Were particular regions of genomes (i.e. the ends of linear chromosomes) typically missing? Or is the incompleteness due to areas of low-quality reads within particular regions of particular genomes? If the latter, were biases seen (certain regions of poor quality in all genomes) or were low-coverage areas distributed randomly among taxa? How much missing data was there within the alignments used for phylogenetic inference?

2. Models of mt gene transcription suggest that in other metazoans transcription is bounded ("punctuated") by tRNAs. Hydrozoan mt genomes code for only two tRNAs, so the authors suggest that the recovery of a total of 6 poly- and monocistronic transcripts from the RADseq data implies that there must be additional punctuation sites that are specified by motifs other than tRNAs. The hypotheses proposed are consistent with the data presented here, but are they consistent with results obtained from RADseq data of other metazoans? In taxa that have a full complement of tRNAs in their mt genomes, does RADseq only recover transcripts (mono- or polycistronic) that are bounded by tRNAs? Or are polycistronic precursors also found? What about other taxa (Anthozoans, sponges) that lack most tRNAs? Seems like there should be enough RADseq data out there to support the model further by confirming (1) that taxa with lots of tRNAs produce many, short transcripts, and (2) other non-Hydroidolinan taxa with few tRNAs produce more transcripts than expected if punctuation were only to occur at tRNAs.

Additional comments

Mostly minor comments that don't really fit into the previous categories:

97: "barcoding approach" is not really the appropriate term for single-locus systematics studies; barcoding typically seeks to delimit species rather than to understand their phylogenetic relationships

102: See also Foox et al 2015 Mitochondrial DNA

137: Table S1 doesn't seem to include "maps of long PCR amplifications" as stated in the text

225: What is SRR871528? A different sample? Different sequencing run? Different database? Table 1 doesn't define what these alphanumeric codes signify.

230-231: Are these GC values the ranges of the average GC contents for genes or for taxa? For example, was 13.1% the lowest GC content observed for a gene (compared to all genes in all taxa), or the lowest GC content observed within a taxon (averaged across all genes)?

244-246: How does this conclusion that there is no compositional bias among taxa square with the very wide range of GC values reported in Table 2, and with the statement (230) that GC content was variable between and within classes?

285: I would have thought tRNAs would be excised rather than incised...

293: aplanatulan? Or aplanulatan?

311: The wording "The presence of gaps within genes in the mtDNA" is not clear. What is meant is that transcripts for some genes were split among different contigs.

329: "This is supported by..." - not clear what "this" refers to.

358-360: repetitive

357-363: Does the statement "Filifera IV was found to include..." refer to the results presented here, or to previous studies? Gonoproxima is not indicated on the tree. It's also not clear how the analyses here support the last sentence of this paragraph.

In conclusions (381) it syas that no new gene orders were found, but wasn't it stated earlier that the Boreohydra arrangement is new?

387-389 is missing some verbs....

·

Basic reporting

This manuscript reconstructs the phylogeny within Hydroidolina, the larger subclass of Hydrozoa, based on near-complete mitochondrial genome sequences. Results show that Siphonophorae is the first diverging clade, sister to a clade comprising Leptothecata + Filifera (III + IV) and Aplanulata + Capitata + Filifera (I + II). The authors investigate mitochondrial gene order and propose a mechanism for mt mRNA expression based on gene order and RNA-seq data. Overall, I find the study to be interesting, utilising a wide variety of new and recent data to infer hydrozoan evolution and mechanism of gene expression. Following are some suggestions to help improve the manuscript further.

Experimental design

No Comments

Validity of the findings

No Comments

Additional comments

The phylogeny based on mitochondrial sequences, even with the complete mt genome, is only part of the story. Despite this being the largest dataset for Hydroidolina, the nuclear genome remains largely unsampled. With 'the ever-decreasing price of sequencing', the authors should comment on whether the nuclear genome could clarify/correct the inferences much more. We know that for some other cnidarians, mitochondrial sequences can be problematic (Kitahara et al. 2014, PLoS ONE). There are some rather long branches (e.g. Filifera I, II) that hint at the possibility of similar issues.

Lines 73–74: The distinction between citations and taxon authorities is unclear. Aplanulata, for instance, should be Aplanulata Collins, Winkelman, Hadrys & Schierwater, 2005. Perhaps state the authority for each taxon, and move and consolidate the citations at the end of the sentence.

The authors mention that their multiplexing strategy is new, but it is not clear which part of their protocol is novel. Labelled ('barcode') primers have been used for some time, and the introduction of double indexing by Illumina means that the combination consequently results in quadruple tagging. Note that Faircloth et al. has also developed this over the last two years for a range of applications.

Lines 189–193: It is not clear here how the best models obtained from the jModelTest and ProtTest outputs were used. Are they applied for each partition or the entire matrix?

Lines 231–238: Integrating these data into Figure 1 will be useful. At the moment there is little context.

Line 249: Please label the orders and taxa in Figures S3 and S7, particularly those recovered as clades in the other analyses. This will help to highlight the discrepancies due to the rRNA data.

Lines 278–280: 'Inter-genic regions (IGRs) longer than 10 bp... might be residues left from the translocations of trnW and trnM...' The authors explained that there are IGRs in the vicinity of mt-tRNAs (line 330), so they could be residues from the translocations, but it's not clear all of them are so. Could the loss of genes (e.g. orf314 and polB) leave these IGRs as well?

Line 294: 'region'

Lines 314–317: 'The absence of any reads between these gene clusters... neither appears to be the result of insufficient read depth nor is it easily explained by the highly transient nature of the polycistronic precursor (pre-mRNA) transcript'. How is this inferred? From the number of reads? There is very little explanation on the coverage of the sequencing runs to build this case.

Fig. 2: 'Hydroidolina'

Fig. 3: 'posterior probabilities'

Fig. 4: Not integrated into the text.

Table 3: The monophyly of the taxa listed here should be evaluated using hypothesis testing methods, e.g. KH (Kishino & Hasegawa 1989) and approximately unbiased (Shimodaira 2002) tests. Relatively high posterior probabilities and bootstrap may not necessarily discount alternative topologies. In addition, it would be useful to summarise some of the morphological character states (already explained in the text) for each of the taxon. This will show clearly the apomorphies that are supported/informative and those that are not.

Table S2: The authors need to indicate here how the Iss and p-values are to be interpreted for suggesting that the rRNA data are saturated while the rest are not.

---

## Round 0.2 · accepted · Accept

Congratulations on a great contribution.